# Domain sliding of two *Staphylococcus aureus* N-acetylglucosaminidases enables their substrate-binding prior to its catalysis

Sara Pintar[1,2], Jure Borišek[3], Aleksandra Usenik[1,2], Andrej Perdih[3] & Dušan Turk[1,2 ✉]

To achieve productive binding, enzymes and substrates must align their geometries to complement each other along an entire substrate binding site, which may require enzyme flexibility. In pursuit of novel drug targets for the human pathogen *S. aureus*, we studied peptidoglycan N-acetylglucosaminidases, whose structures are composed of two domains forming a V-shaped active site cleft. Combined insights from crystal structures supported by site-directed mutagenesis, modeling, and molecular dynamics enabled us to elucidate the substrate binding mechanism of SagB and AtlA-gl. This mechanism requires domain sliding from the open form observed in their crystal structures, leading to polysaccharide substrate binding in the closed form, which can enzymatically process the bound substrate. We suggest that these two hydrolases must exhibit unusual extents of flexibility to cleave the rigid structure of a bacterial cell wall.

[1] Department of Biochemistry, Molecular and Structural Biology, Jožef Stefan Institute, Jamova cesta 39, 1000 Ljubljana, Slovenia. [2] Centre of Excellence for Integrated Approaches in Chemistry and Biology of Proteins, Jamova cesta 39, 1000 Ljubljana, Slovenia. [3] Theory Department, National Institute of Chemistry, Hajdrihova 19, 1001 Ljubljana, Slovenia. ✉email: dusan.turk@ijs.si

The mechanistic understanding of enzymatic reactions has been pursued since the beginnings of biochemistry. It started with the rigid lock and key model of Fischer in 1894[1]. However, over time, the dynamic nature of proteins became increasingly important. When studying protein synthesis in 1958, Koshland proposed the induced fit model, which posits that substrate binding can cause an enzyme conformational change, thereby bringing the catalytic groups into the precise position required for the reaction to occur[2]. In recent decades, the importance of the dynamic nature of proteins for their physiological and biochemical properties has gained momentum[3,4]. To overcome limitations associated with the induced fit model, protein flexibility in solution has been accounted for through the conformational selection mechanism, which suggests that enzymes adopt a range of conformations that can bind a substrate and that binding then shifts the equilibrium in favor of the bound conformation[5]. It has been suggested that both theories are not mutually exclusive, as there have been an increasing number of cases where conformational selection was shown to be followed by conformational adjustment[6].

Experimentally determined structures and computational approaches have been used to gain insights into the conformational landscape of biological macromolecules. The majority of protein structures at various atomic resolutions have been determined by protein crystallography, yet crystal structures provide very limited insights into protein flexibility, and was very aptly pointed out by Gutteridge and Thornton, there is an inherent bias towards identifying small motions due to the restraints that this method imposes[7]. Taylor et al. found the most common rotation angle was 15° with angles above 30° rarely observed[8]. When analyzing the entire PDB for multiple structures of the same protein before and after ligand binding, Amemiya et al. considered large motions as the displacement of $C_\alpha$ atoms by more than 1.0 Å[9]. In contrast to crystallography, NMR and cryo-EM provide insights into conformational dynamics within a single data set; however, their accuracy often defies unambiguous interpretation of conformational ensembles. Contrary to experimental methods, computational approaches are not restrained by a data set; however, they suffer from restrictions imposed by the unavoidable simplifications of molecular models coupled with limited computational resources. This suggests that combination of experimental and computational approaches appears most suitable for these studies[10].

In spite of the limited insights into molecular flexibility, trapping conformational states in a crystal has still enabled us to obtain insights into the dynamic properties of enzymes. In our quest for potential drug targets, we analyzed crystal structures of the glycoside hydrolase 73 family (GH73)[11] of enzymes from the human pathogen *S. aureus* and came across a case of exceptional mobility within a pair of enzymes that extended our understanding of conformational flexibility and shed light on their biological function. GH73 enzymes have been shown to be involved in cell division[12–14] and flagellar rod formation[15]. Their substrate is a component of the bacterial cell wall, a carbohydrate polymer composed of alternating N-acetylglucosamine (NAG) and N-acetylmuramic acid (NAM). We recently determined the crystal structures of a homologous protein, autolysin E (AtlE) in complex with substrate fragments, which we used together with other available structural data on lysozymes to build a model of (NAG-NAM)$_3$ substrate binding to AtlE[16]. We have also analyzed the amino acids that are important for the reaction to transpire and identified E138 as the catalytic general acid and Y224-A225-S226-D227 as the motif responsible for substrate binding[17].

Here, we present two crystal structures of GH73 enzymes from *S. aureus*, including glucosaminidase B (SagB) and the glucosaminidase

part of major bifunctional autolysin (AtlA-gl). Inactivation of SagB instigates slower culture growth, cell morphological defects as well as the perturbed excretion of cytoplasmic proteins[18,19]. AtlA is a two domain protein that is post-translationally cleaved into two proteins, each with its own enzymatic specificity, N-acetylglucosaminidase and amidase[20]. AtlA has been shown to be localized at the septal region and to be involved in cell division[14,21]. We combined the understanding of substrate binding obtained from our previous AtlE studies[16,17] with insights obtained from the SagB and AtlA-gl structures. The analysis of substrate binding sites revealed that for the active site residues to enable the enzymatic reaction to proceed, a substantial conformational change has to take place. This was corroborated by point mutations and molecular dynamics simulations.

## Results

**Description of AtlA-gl and SagB crystal structures**. SagB crystallized in the $P6_122$ space group with two molecules per asymmetric unit (Table 1). The structure of SagB is well-resolved based on electron density maps along the entire chain with the exception of the N36 to A41 region in both molecules, as indicated by the Ramachandran outliers and higher atomic *B*-values, as well as a few other side chains. The crystal structure also contains eight PEG molecules (PG4), two chloride ions and one sodium ion. The latter is positioned at the crystallographic two-fold axis. The two molecules superimpose with a root-mean-square-deviation (RMSD) of 0.46 Å over all $C_\alpha$ atoms. The superimposition parameters slightly deviate from a proper noncrystallographic symmetry dimer. The structure of SagB shares the topological layout and heart-like shape of AtlE (4PIA)[16]. it is composed of right (R) and left (L) domains divided into core and lobe regions (Fig. 1). The catalytic residue E121 is positioned at the C-terminus of helix α10. The R-domain comprises residues from A1 to D71 and S187 to K250, whereas the L-domain comprises residues from L72 to S186. The two domains in molecule A superimpose onto their counterparts in molecule B with similar RMSDs (0.46 and 0.38 Å) indicating equivalent domain packing.

AtlA-gl crystallized in the $P6_422$ space group with one molecule per asymmetric unit (Table 1). In spite of the weak diffraction data, its structure is well-resolved based on the electron density maps with the exception of the two N-terminal amino acid residues and the A128-N140 loop. The crystallographic R-factor of 0.29 reflects the irreproducible crystal, its small size and the low intensity of the diffraction data. Similar to SagB, the AtlA-gl structure also shares the topological layout of AtlE (Fig. 1). Its R-domain comprises the region from A1 to D68 and A190 to K244, whereas the L-domain comprises from Q69 to K176. In contrast to the R-lobe, the L-lobe is smaller and partially disordered.

**Comparison of the AtlA-gl, SagB, and AtlE structures**. The failure of molecular replacement was the first indication that the SagB structure must differ from the AtlE structure in spite of their high sequence similarity. When the structure was determined, it became evident that the two structures share the same fold; however the L-domains and R-domains are positioned differently relative to one another (Fig. 1). To quantify the differences, the structures were superimposed as a whole and in parts with the FatCat[22] interface imbedded in MAIN[23]. Superimposition of the complete structures yielded an RMSD of 3.4 Å over 195 residues with 28% sequence identity; in contrast, superimposition of the two parts corresponding to the R-domains and L-domains yielded an RMSD of 2.2 Å over 104 matched residues with 28% identity and an RMSD of 1.7 Å over 98 matching residues with the 36% identity for the R-domain and L-domain, respectively.

**Table 1 Data collection and refinement statistics.**

|  | AtlA-gl | SagB | SagB_SeMet_remote |
|---|---|---|---|
| Data collection |  |  |  |
| Space group | P 64 2 2 (181) | P 61 2 2 (178) | P 61 2 2 (178) |
| Cell dimensions |  |  |  |
| $a, b, c$ (Å) | 110.17, 110.17, 92.39 | 151.77, 151.77, 122.51 | 152.78, 152.78, 123.83 |
| $\alpha, \beta, \gamma$ (°) | 90, 90, 120 | 90, 90, 120 | 90, 90, 120 |
| Wavelength (Å) | 0.9184 | 0.8943 | 0.9770 |
| Resolution (Å) | 42.39–2.28 (2.41–2.28) | 47.66–2.03 (2.15–2.03) | 48.10–2.19 (2.32–2.19) |
| $R_{measured}$ (%) | 9.1 (162.6) | 9.1 (105.0) | 17.5 (192.7) |
| $I/\sigma I$ | 25.31 (1.96) | 27.07 (2.02) | 17.06 (1.83) |
| Completness (%) | 99.8 (99.3) | 99.7 (98.5) | 99.8 (99.0) |
| Multiciplicity | 21.1 | 18.3 | 21.2 |
| CC(1/2) | 100.0 (63.1) | 99.9 (70.4) | 99.9 (62.8) |
| ISa | 24.95 | 31.33 | 24.85 |
| Refinement |  |  |  |
| Resolution (Å) | 36.06–2.50 (2.54–2.50) | 19.95–2.03 (2.07–2.03) |  |
| No. reflections | 11943 (578) | 53814 (2544) |  |
| $R_{work}/R_{kick}$[1] | 0.295/0.334 | 0.212/0.220 |  |
| No. atoms |  |  |  |
| Protein | 2261 | 4046 |  |
| Ligand/ion | 3 | 107 |  |
| Water | 162 | 548 |  |
| B-factors |  |  |  |
| Protein | 68.1 | 40.2 |  |
| Ligand/ion | 68.1 | 66.6 |  |
| Water | 66.6 | 66.2 |  |
| R.m.s. deviations |  |  |  |
| Bond lengths (Å) | 0.79 | 0.78 |  |
| Bond angles (°) | 1.08 | 0.78 |  |

[1]The idea behind $R_{kick}$ is similar to the $R_{free}$, however, it is calculated from the structure factors of the kicked model against WORK set of data and not from structure factors of unperturbed model against TEST part of data. This corrects the conceptual problem of $R_{free}$. Namely, the claim that TEST set used in cross validation maximum likelihood refinement target function as source of structure independent information is false because errors in models under refinement are not randomly distributed[55].

Moreover, the angle of rotation of the SagB structure super-imposed on the AtlE R-domain with the SagB structure superimposed onto the AtlE L-domain is 31° with a 3.5 Å screw translational component. Thus, the packing of the R-domains and L-domains in the SagB structure differs from their packing in the AtlE structure.

The AtlA-gl structure was solved by molecular replacement using the SagB structure as the search model. A similar comparison with the AtlE structure as described above for the SagB structure yielded an RMSD of 4.0 Å over 199 matching residues with 14% identity, whereas the superimposition of the R-domains and L-domains yielded RMSDs of 2.2 and 1.7 Å over 104 and 99 matching residues, respectively. The angle of rotation of the AtlA-gl structure superimposed on the R-domains and L-domains is 39° and resulted in a screw translational component of 4.6 Å. Thus, the packing of the R-domains and L-domains in the AtlA-gl structure differs from their packing in the AtlE structure even more than in the SagB structure.

The positioning of the conserved core helices is equivalent to that of AtlE in seven other known GH73 crystal structures[16], the only two structures that substantially deviate are SagB and Atl-gl (Supplementary Fig. 1). Apart from the R-domain and L-domain packing, the major difference among the three structures is in the helix region connecting the R-lobes and L-lobes marked with a rectangle in Figs. 1, 2. In the AtlA-gl structure, the 4th turn of helix α13 is stretched out compared to the slightly longer equivalent helix α15 in SagB and the even longer helix α12 in AtlE. The equivalent helix in the AtlE structure is distorted after the fourth turn with the aromatic residue Y201 slightly exposed. The α15 helix in SagB is almost a whole turn shorter and shares the same minor widening in the last turn of the helix at position F185.

**Substrate binding**. As a consequence of the different positioning of the L-domain and R-domain substrate binding clefts in SagB and AtlA-gl, structures are much wider in comparison to the AtlE structure. This is reflected in the distances between the catalytic glutamate residue $C_\alpha$ atoms (E121 in SagB, E116 in AtlA, and E138 in AtlE) from the L-domain and the $C_\alpha$ atoms from the conserved YA(S/T)D region (Y209-D212 in SagB, Y214-D217 in AtlA-gl, and Y224-D227 in AtlE) positioned in the opposite side in the R-domain (Fig. 2). The shortest distances between the catalytic glutamates and YA(S/T)D regions were measured in the AtlE structure and are in the range from 10.8 to 12.8 Å, followed by the distances from the SagB structure in the range from 17.6 to 20.5 Å, and the distances from AtlA-gl structure in the range from 17.7 to 21.8 Å. Hence, the substrate-binding clefts of AtlA-gl and SagB, with an 8 Å difference, are wider than their equivalent in the AtlE structure.

To gain insights into the possible (NAG-NAM)₃ substrate binding geometry in the AtlA-gl and SagB structures, we used the binding geometry previously established for the AtlE structure[16]. When bound to AtlE, the substrate was in contact with the catalytic glutamate residue in the L-domain and the YA(S/T)D region in the R-domain in the active site cleft. The only way to conserve these previously identified contacts in the L-domains and R-domains was to independently superimpose the (NAG-NAM)₃ AtlE-substrate complex to each domain independently. This resulted in two putative substrate binding geometries (Fig. 3). When the L-domain pairs were superimposed, the (NAG-NAM)₃ substrate retained its position over the catalytic glutamates, and when the R-domains were superimposed, the substrate retained the contacts with the YA(S/T)D binding region. This suggests that a conformational change has to occur to bring those two

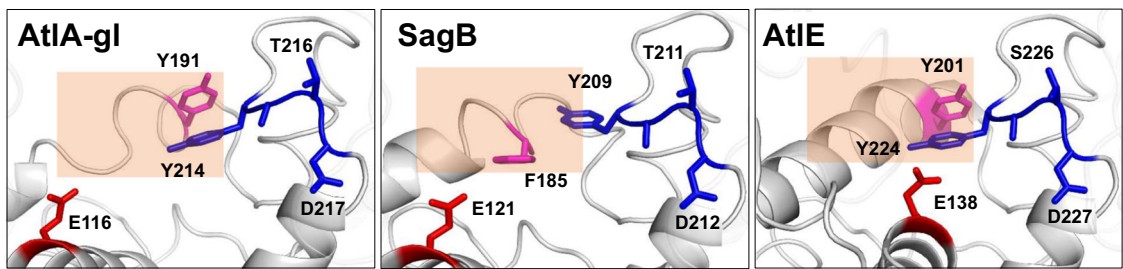

**Fig. 1 Structures of AtlA-gl, SagB and AtlE.** The left column shows the ribbon presentation of the structures, whereas the right column shows topology diagrams. The highly conserved helices in the lysozyme-type fold enzymes are depicted in blue and the other core secondary structure elements are shown in cyan. The L-lobe secondary structure elements are shown in green and the R-lobe elements are in yellow. The active site glutamate is marked in red. An orange rectangle marks the region of differences between helices α13 in AtlA-gl, α15 in SagB, and α12 in AtlE.

**Fig. 2 Conserved amino acid residues in the AtlA-gl, SagB, and AtlE substrate binding sites.** In the white ribbon structure, the catalytic glutamates (E116, E121, and E138) are shown as red sticks, the YA(S/T)D regions as blue sticks, and the conserved aromatic residues (Y191, F185, and Y201) as pink sticks. Note that S226 has a double conformation in the AtlE crystal.

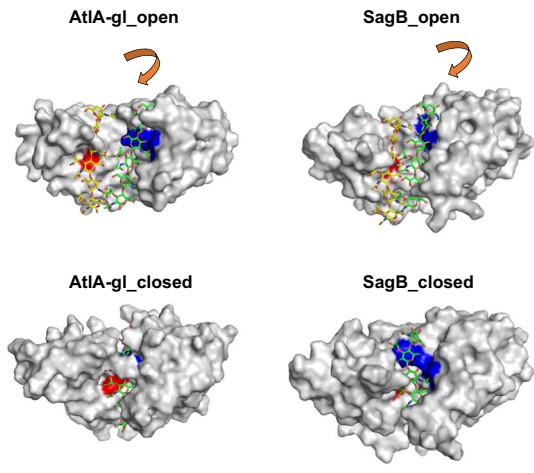

**Fig. 3 The (NAG-NAM)₃ substrate binding models of the AtlA-gl and SagB open and closed forms.** The color coding of the surface is the same as used in Fig. 2. The top two structures include (NAG-NAM)₃ shown as a stick model with carbon bound to the L-domains and R-domains colored yellow and green, respectively. The bottom images represent the closed structures with bound (NAG-NAM)₃ substrate. The NAM(−2) and NAG(−1) residues in the pair of substrate models attached to the L-domain exhibit clashes with the features in the SagB (A124 and K127) and AtlA-gl (Y152) structures, which can be resolved with minor conformational adjustments. The arrows indicate the direction of domain rotations.

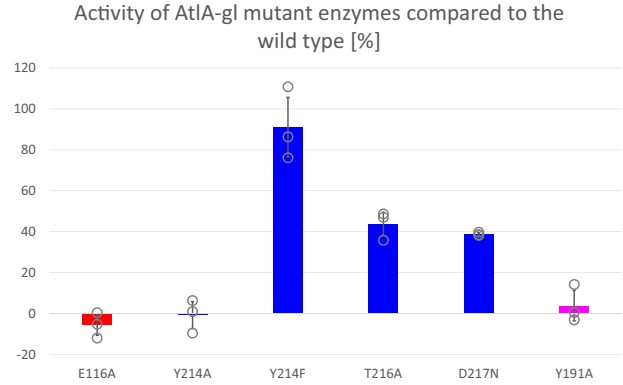

**Fig. 4 The effects of the point mutations on the AtlA-gl substrate binding site residues on the peptidoglycan-degrading ability.** The AtlE mutations were described by Borišek et al.[17].

regions together. We have previously shown that AtlA-gl can cleave (NAG-NAM)₂ tetrasaccharide[16], similar to AtlE, which indicates that despite the large distance between the substrate binding region and the catalytic site, a single glycan strand is a substrate for this enzyme. This suggests that the AtlA-gl and SagB structures were observed in the open form compared to the AtlE structure, which was observed in the closed form.

To identify the equivalent residues in AtlA-gl involved in substrate binding as in AtlE[17], we introduced point mutations that either targeted the catalytic Glu directly or the substrate binding YA(S/T)D region (Fig. 4). An analogous substrate degradation assay was not functional for SagB. All produced AtlA-gl mutants were correctly folded as confirmed by CD spectra. Mutation of the catalytic glutamate E116 to alanine rendered enzymes inactive, as has been shown previously for several enzymes in this family, including AtlE[17], AtlWM[24], AcmA[25], LytB[26,27], SpFlgJ[28], StFlgJ[29], Auto[30], TM0633[31], and AcpCD[32]. The mutation of the tyrosine residue Y214 in the YATD region to alanine severely affected enzyme activity, whereas mutation of Y214 to phenylalanine completely restored the substrate degrading ability. Similar observations were previously established for AtlE[17], as well as for four other GH73 enzymes[24,25,28,32,33], demonstrating that an aromatic residue is required at this position. The T216A and D217N mutants exhibited a marked decrease in activity. Similar observations were made for the equivalent S226A and D227A mutants in AtlE[17]. We were unable to obtain a soluble D217A mutant from AtlA-gl, indicating that D217 is important for the structural stability of the region and protein folding. Mutation of the conserved aromatic residue Y191A positioned within the helix connecting the L-domains and R-domains resulted in an inactive AtlA-gl, which was equivalent to mutation of the equivalent residue Y201A in AtlE, which also resulted in an inactive enzyme. From these results, we concluded that the catalytic glutamate and the YA(S/T)D region are both involved in substrate binding during hydrolysis. Hence, the structures must adopt closed forms to perform the reaction.

To gain insights into the structural flexibility of the two proteins closed form models were constructed. Both were modeled using the AtlE structure as the template. Simple models were created by superimposition of each domain of SagB and AtlA-gl to corresponding domain of AtlE followed by minimization of the combined models (Fig. 1). The models obtained by superimposition resulted only in minor side chain clashes: 24 non-hydrogen atoms from four residues of the L-domain and 23 non-hydrogen atoms from five residues of D-domains of AtlA-gl were closer than 2.2 Å (the distance used by PDB structure validation tools to indicate possible false covalent bond interactions), whereas for the SagB 30 non-hydrogen atoms from five residues of the L-domain and 34 non-hydrogen atoms form five residues of the D-domain were closer than 2.2 Å. These clashes were resolved by minimization. These simple models were further adjusted to match the chain trace of the α12 helix in the AtlE template (marked with an orange rectangle in Fig. 1, see Methods section for further details).

According to DynDom[34,35] analyses the transitions between open and closed forms of AtlA-gl and SagB were classified as hinge motion with a 35.3° rotation angle, −0.7 Å translation and 97.3% closure for AtlA-gl (Supplementary Movie 1) and 29.4° rotation angle, −1.1 Å translation and 83.3% closure for SagB (Supplementary Movie 2). According to the structural flexibility analysis, structures with such high rotation angles are rare[8]. Due to the specifics of the structural change, it seems more appropriate to describe the domain movement as sliding domains along their interface than rotation. The extent of changes during transition between the open and closed forms was analyzed by counting the unique contacts between residues and their RMSDs (for details see Methods section). Visualization of the changes in inter-residue contacts revealed their location (shown in red in Supplementary Fig. 2) along the two domain interfaces of the AtlA-gl and SagB structures. The AtlA-gl pair of open and closed forms has 46 and 76 inter-residue specific interactions, respectively, and an RMSD of 4.5 Å, whereas the SagB pair of open and closed forms has 96 and 100 inter-residue specific interactions, respectively, and an RMSD of 3.6 Å. In comparison, the pair of SagB structures related by non-crystallographic symmetry has 70 and 66 inter-residue specific interactions and an RMSD of 0.6 Å, whereas the pair of T and R states of allosterically regulated L-Lactate dehydrogenase has 204 and 228 unique interactions, respectively, and an RMSD of 1.9 Å. The differences in the RMSD deviations between the pairs of structures in different forms and the similarity of the number of differences between the inter-residue specific interactions suggest that the differences between the open and closed forms of the AtlA-gl and SagB structures are likely larger than modeled. To show that also the closed forms of

both enzymes can accommodate peptidyl links attached to polysaccharides we built the models using the bound (NAG-NAM)$_3$ substrate with added L-Ala-D-iso-Gln-L-Lys part of the peptide stem (Supplementary Fig. 3).

**Molecular simulations of AtlA-gl and SagB.** To investigate the stability of the crystal structures and the models of the closed forms of the AtlA-gl and SagB structures, crystal structures and models were subjected to several classical molecular dynamics (MD) simulations in the time span of 500 ns for each performed simulation.

First, we performed MD simulations on the AtlA-gl and SagB crystal structures. Their MD-generated conformations exhibited RMSD values of 2.56 ± 0.32 Å (AtlA-gl) and 1.99 ± 0.41 Å (SagB). The distances between the C$_\alpha$ atoms from the catalytic glutamate and alanine in the YATD region were in the range from 20.3 ± 0.9 Å and 14.9 ± 2.0 Å for AtlA-gl and SagB, respectively, which are close to the values observed in the starting structures. The residues from the lobe parts displayed substantial motion indicated by the larger thickness and yellow/red color of the chain trace (Supplementary Fig. 4; Supplementary Table 1). Their trajectories did not display any significant tendencies of transition towards their closed forms.

We then simulated the simple closed form models of both enzymes. The MD-generated conformations were highly unstable as indicated by the substantially higher structural RMSD values of 4.76 ± 0.71 Å (AtlA-gl) and 3.52 ± 0.94 Å (SagB) compared to the other MD simulations (Supplementary Fig. 5; Supplementary Table 1). There was an increase in the interdomain distance between the catalytic glutamate and the YATD region of approximately 4 Å in AtlA-gl and 3 Å in SagB when compared with the starting structure. This suggested that both simple closed forms exhibited a tendency for their transition towards the open form. Although considerably long, a 500-ns MD simulation was not long enough to observe the full conformational transition.

Next, we simulated the closed forms with the rebuilt helices. During the MD simulations, both models remained in their closed forms with the observed average distances were 12.2 ± 0.87 Å (AtlA-gl) and 12.9 ± 0.78 Å (SagB) (Supplementary Fig. 4B). This suggested that the adjustment of the length of the domain-connecting helix indeed contributed to the stability of the closed forms of AtlA-gl and SagB.

Further, we simulated the closed forms of AtlA-gl and SagB with rebuilt helices and the bound substrate (NAG-NAM)$_3$. In both simulations, the geometries remained stable (Supplementary Fig. 4; Supplementary Table 1). Compared with the simulation of the closed forms without the bound substrate, the average interdomain distances were slightly shorter in the case of AtlA-gl (10.4 ± 0.32 Å) and remained about the same as in SagB (13.0 ± 0.78 Å) (Supplementary Fig. 4).

Taken together, the molecular simulations confirmed the stability of the rebuilt closed forms and their complexes with substrate models. Furthermore, our analysis suggested that the interdomain conformational flexibility of AtlA-gl and SagB is accompanied by a partial helix-folding/unfolding mechanism.

## Discussion

SagB and AtlA-gl crystallized in hexagonal crystals in unit cells of different size and packing. SagB crystallized with two and AtlA-gl with one molecule per asymmetric unit. In contrast to the crystal structure of AtlE and related N-acetylglucosaminidases, which were all observed in the closed form, these two proteins crystallized in the open form. The two independent crystal structures of the two distinct proteins exclude the possibility that these results are both crystallization artifacts. Hence, the observed open forms

must reflect the real structural dynamics of these two bacterial N-acetylglucosaminidases.

Comparison of the SagB, AtlA-gl, and AtlE crystal structures revealed that the fold of the L-domains and R-domains of all three proteins is the same; however, the relative position of the domains exhibit unusually large differences. With respect to the closed form of the AtlE structure, the L-domains and R-domains in the SagB and AtlA-gl structures are rotated away from each other, resulting in a very wide substrate binding grove. Analysis of the substrate binding geometry together with site-directed mutagenesis identified residues in the L-domain and R-domain that are essential for the reaction to transpire. Hence, for AtlA-gl and SagB to cleave a single glycan strand, both of their domains have to slide together into a closed conformation as observed in the AtlE structure. The suggested conformational changes were corroborated by MD simulations.

Analysis of the specific inter-residue contacts has shown that they are located along the two domain interfaces; in the open forms, the contacts are located bellow the substrate binding site, whereas in the closed forms, they were positioned higher in the substrate binding region. AtlE appears to be able to bind the substrate in its closed form with minimal adjustments[17]. In contrast, the closed forms of SagB and even more so that of AtlA-gl embrace the substrate (Fig. 3), meaning that the substrate cannot bind to the active site without the transition of the enzymes into their open forms, indicating that closing occurs after substrate binding. In Rnase A, for example, it was shown that both open and closed forms exist in solution[36]. Similarly, open and close conformations associated with loop and helix flexibility were found in the substrate-free enzyme structures of adenylate kinase[37]. These works showed that the open forms of the AtlA-gl and SagB structures are not an exception. However, the kind and extent of flexibility of N-acetylglucosaminidases associated with domain sliding and helix transition described here surpass the range of expectations. Even our 500-ns long molecular dynamic simulations could only confirm the stability of the observed and modeled open and closed forms, but not reproduce their transitions.

The question here is, why are these molecular gymnastics necessary? It is common knowledge that enzymes catalyze chemical reactions by aligning the reactive groups in an optimal position for efficient chemistry to take place. Fersht proposed that as the enzyme is more flexible than the substrate, its distortion can be larger[38]. The S. aureus cell wall has been found to be more rigid than that of E. coli or B. subtillis[39], and even further, glycan strands were shown to be less flexible than the peptide portions of peptidoglycan[40]. Moreover, it was also demonstrated that S. aureus cells deficient in GH73 enzymes have increased surface stiffness, with SagB inactivation having the largest effect, as well as the important contributions of other enzymes, including AtlA-gl and AtlE[18]. These findings further indicate that the substrates of these enzymes are particularly rigid and therefore require corresponding flexibility by the enzymes processing them. There are reports of conformational flexibility in enzymes that synthesize, hydrolyze or bind peptidoglycan glycan strands, and they range from altered relative positioning of different domains connected by a linker region[41–44] to loop flexibility[45] and intradomain movement[46]. Analysis of PDB structures with and without ligands exposed that among enzymes, only transferases and hydrolases exhibit domain motion; however, the extent of motion observed for hydrolases was smaller (below 1 Å RMSD)[47], suggesting that the chemical reaction of linked chemical groups favors solvent exclusion, whereas hydrolysis requires solvent, explaining the smaller structural changes in hydrolases. The extent and kinds of movement observed in our structures however hints in the other direction. Taken together, our structural analysis suggests that these two hydrolases must have evolved an unusual extent of flexibility to cleave the targeted glycosidic bonds in rigid bacterial cell walls. It

is not chemistry alone that requires adaption of flexibility, but also the large and rigid substrate which needs to be torn. Bailey et al. have shown that the newly formed peptidoglycan at the septum is stiffer and proposed that the subsequent drop in rigidity during division process is likely due to the peptidoglycan hydrolase activity[48]. This coincides with AtlA localization at the septal region and its proposed role in cell division[14,21] and their ability to cleave highly cross linked *S. aureus* peptidoglycan[19]. Superimposition of other eight structures from GH73 family deposited in PDB provided no evidence that any of them were crystallized in other than the closed form (Supplementary Fig. 1). Hence, the insight into AtlA and SagB may shed a light on synthesis of peptidoglycan in *S. aureus*, which is inserted only during cell division at the septum[49]. Taken together, we can speculate that SagB and AtlA-gl have a role in cell division which involves reducing the rigidity of the peptidoglycan.

## Methods

**Protein production and crystallization**. Protein coding sequences were cloned into the pMCSG7 vector using LIC procedures[50] and KOD Hot Start Polymerase (Invitrogen). *S. aureus* Mu50 genomic DNA (ATCC) was used as the template with the primers listed in Supplementary Table 2. Proteins were produced in *E. coli* BL21DE3 grown in ZYM-5052 autoinduction media for 5 h at 37 °C and 18 h at 20 °C. Cells were resuspended in buffer A (30 mM Tris-HCl, pH 7.5, and 400 mM NaCl) with 1 mg ml$^{-1}$ lysozyme and by sonication. The lysate was cleared by centrifugation. Proteins were purified by Ni-NTA using buffer A with 10 mM imidazole for binding and 300 mM imidazole for elution, followed by SEC (HiPrep 16/60 Sephacryl S-200 HR (GE Healthcare)) with buffer A. The His tag was cleaved off with TEV protease and removed by Ni-NTA. Mutant enzymes were produced by IPTG induction in LB medium at an OD$_{600}$ of approximately 0.6 and purified by the same procedure as the wild type enzymes. CD spectra confirmed mutant proteins assumed the same fold as the wild type proteins.

For crystallization, SagB was dialyzed into 20 mM citrate, pH 5, with 100 mM NaCl and AtlA-gl into 20 mM HEPES, pH 7.5 with 100 mM NaCl and concentrated to 13 mg ml$^{-1}$ and 38 mg ml$^{-1}$, respectively. SagB was crystallized in 0.1 M Na$_3$citrate, pH 4.75 with 47% PEG 200 and AtlA-glu in 2 M (NH$_4$)$_2$SO$_4$. AtlA-gl crystallized in the form of several crystals that grew from a single point. Crystals were small and not reproducible. Both proteins were crystallized using the sitting drop at 20 °C.

**Activity assay**. Peptidoglycan was purified and dyed and the assay was conducted as described previously[17]. The buffers used in the activity assay were 100 mM Tris, pH 8, for AtlA-gl and 100 mM MES, pH 5.5, for AtlE. AtlA-gl and AtlE activity assays were performed with 5 μM enzymes at 25 °C and 600 rpm.

**Protein structure determination**. Diffraction data sets for SagB and AtlA-gl were collected at 100 K using synchrotron radiation at the BESSY 14.1. and BESSY 14.3. beamlines at Helmholtz-Zentrum Berlin[51]. Indexing, integrating and scaling were performed with the XDS software package[52]. Native SagB crystals diffracted to 2.0 Å resolution, whereas SeMet crystals diffracted to 2.6 Å. The SagB structure was solved using an SeMet derivative data set collected at a remote wavelength of 0.977031 Å. Initial chain trace was performed by ShelX[53]. The AtlA-gl crystal diffracted to 2.3 Å. The phase problem for AtlA-gl was solved by molecular replacement of the SagB structure using PHASER[54]. Model building, refinement, and structure validation for SagB and AtlA-gl were performed with MAIN[23]. During refinement, the Free Kick maximum likelihood target used all diffraction data for both working and testing simultaneously[55].

**Construction and analysis of closed structures**. The closed form models of both structures were constructed in two steps. First, we created the models by superimposition of the R-domains and L-domains on the AtlE structure. L-domains contained residues from sequence ranges from Q69 to K176 and L72 to S186 from AtlA-gl and SagB, respectively, whereas the R-domains contained the omitted parts. Then the broken link and side chain clashes were adjusted by energy minimization. Visual inspection of the superimposed models indicated the position of the pivoting point proximal to residues I89 in AtlE, L67 in AtlA-gl, and L106 in SagB. We called the generated models simple models. These two models were further modified by extension of the domain-connecting helices. The α13 helix in the AtlA-gl closed form model from S175 to K191 and the α15 helix in the SagB closed form model from P169 to S186 were rebuilt to match their chain traces to the α12 helix in the AtlE structure.

The extent of change of the contacts was analyzed by packing analysis of the structures using graph nodes described recently[56]. Here we calculated graphs of intra residue interactions within a 4.6 Å radius using only nonhydrogen atoms and disregarding two neighboring residues in the chain. The connectivity graphs for each pair of molecules were then subtracted from one another so that all matching connections were removed. The resulting differential graphs contained only interactions specific for the form and molecule.

**MD simulations**. The MD calculations were performed using the NAMD molecular modeling suite[57] at the Azman Computing Center at the National Institute of Chemistry, Slovenia. AtlA-gl loop residues 128–140 from the 6FXO crystal structure that were not resolved in the experimental electron density map were constructed using Modeller[58]. Enzymes and substrates were prepared according to our previously published protocol[17]. Briefly, the protonation states of all ionizable amino acid residues were assigned based on the pKa estimated by PROPKA version 3.0[59] at pH = 6.0. All aspartate and glutamate residues were deprotonated, with the exception of catalytic Glu116 and Glu121 from AtlA-gl and SagB, respectively.

The CHARMM-GUI environment[60] was utilized for protein manipulation and construction of the solvated protein-substrate complexes. CHARMM parameter and topology files (version 27)[61,62] were utilized to specify the force field parameters for the amino acid residues comprising the protein. The CHARMM General Force Field (CGenFF)[63] was used to describe the atom types and substrate partial charges[17]. The final systems prepared for the MD simulation was composed of 39,264 and 45,160 atoms for AtlA-gl and SagB.

Both systems were minimized for 10,000 steps with the steepest descent method followed by 10,000 steps of a modified Adopted Basis Newton–Raphson method and a 1-ns MD equilibration run. The production MD trajectory was generated using a leapfrog integration scheme and a 2-fs simulation step using the SHAKE algorithm. A 500-ns MD simulation production run was performed for each model. Conformations were sampled every 1000th step resulting in 25,000 conformations for subsequent analysis.

Visualization and analysis of the geometric parameters for the production MD trajectory were performed using the VMD program[64]. Structures, measured distances and RMSD graphs for all performed systems (AtlA-gl and SagB) are available in Supplementary Figs. 4, 5; Supplementary Table 1.

**Reporting summary**. Further information on research design is available in the Nature Research Reporting Summary linked to this article.

## Data availability

The atomic coordinates and structure factors for AtlA-gl and SagB have been deposited in the Protein Data Bank under accession codes 6FXO and 6FXP, respectively. The source data underlying the plots shown in Fig. 4 are provided in Supplementary Data 1.

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

## Acknowledgements

The authors thank Sandra Gabelli and Marcin Cymborowski for consultancy on AtlA-gl diffraction data processing and acknowledge financial support from the Slovenian Research Agency (research core funding no. P1–0048, P1–0017, P1–0012 and IO-0048, and project no. J1–8152). Data were collected at BMX14.1 and BMX 13.3 (BESSY, Berlin).

## Author contributions

D.T. conceived the project. S.P. and A.U. performed the cloning, protein expression and purification. S.P. performed crystallization and collected the X-ray diffraction data. S.P. and D.T. determined the structures and analyzed them. S.P. performed the activity assays. D.T. constructed and analyzed the closed enzyme models. J.B. and A.P. performed MD simulations and analyzed the results. S.P. and D.T. wrote the manuscript. A.P., A.U., and J.B. edited the manuscript.

## Competing interests

The authors declare no competing interests.
