## [Peer Review File · Communications Biology]

Reviewers' comments:

Reviewer #1 (Remarks to the Author):

The manuscript by Pintar et al describe the structural determination of two glycoside hydrolases from family GH73, SagB and AtIA-gl from *S. aureus*. Comparison of these two structures with AtIE from *S. aureus*, previously reported by the same group, reveals a different conformation for the two domains forming the active site.

Authors propose existence of two different conformations, closed and open, that could rearrange active site in order to perform catalysis. This hypothesis is supported by both molecular dynamics simulations and site directed mutagenesis. The structural analysis is sound and figures clear. A weak point in this manuscript is lack of the in vivo implications for this system, something that limits the impact of the present study.

Main concerns:

1. While structural determination of SagB seems to be OK, there are some doubts about the quality of the AtIA-gl structure as deduced by the statistics presented in the crystallographic Table. The final values for R and R_{free} (I guess this is the meaning of the line named "Rkick", otherwise please explain) are very high for a medium resolution (2.28 Å) structure, and no clear explanation is provided by the authors. Is there any problem in the experimental data related with anisotropy, twinning? Limitation of the experimental data to lower resolution could correct it?.
2. PDB validation reports for the two structures should be add as supplemental material in order to fully asses the quality of the structures. As well as a figure providing details of the electron density quality in the active site.
3. Crystallographic Table presents mistakes and some relevant information is missing, that could indicate a lack of care in preparation of the Table.
 - a. Provide space group names as in the main text.
 - b. Instead of multiplicity values for data sets, some dates appear (?)
 - c. What Rkick means? Is R_{free} ? if not, please explain.
 - d. No values are present in the Table showing quality of the refinement in stereochemistry, Ramachandran, B-factors etc
4. In order to understand how these autolysins could act in vivo a better description of changes in the active site upon rearrangement should be provided.
5. Is there any indication if *S. aureus* autolysins recognize naked peptidoglycan or peptide stems are required? In this case, how peptide stem could be affected by the structural arrangement? Are crosslinked peptidoglycan strands substrate for these autolysins?
6. Is this rearrangement expected for other GH73 family members?

Minor points:

- Line 127. Please provide a supplemental figure showing conservation of such structural elements in GH73 family.
- Line 146. "binding geometry previously established for the AtIE structure" please provide reference.
- Line 151-152: "Figure 3" is doubled.

Reviewer #2 (Remarks to the Author):

The manuscript reported two crystal structures of SagB and AtIA-gl, two peptidoglycan N-acetylglucosaminidases from *S. aureus*. The authors also used modeling and molecular dynamics to

investigate the conformational dynamics of these V-shaped structures in relation to substrate binding and catalytic activity.

Major concerns:

- There is insufficient introduction on AltE and its structure in complex with (NAG-NAM)₃. So it is not easy to follow the definition of "open" and "closed" states for these enzymes. The author can consider adding a scheme that shows the stepwise transition from the "open" form observed in the apo structure to the "closed" form ready for catalysis.

- The description of modeling the "closed" structure of SagB and AtIA-gl by extending the domain-connecting helix is not clear. What extension was added to generate a model that is eventually stable in MD simulation? This may lead to model bias, which should be discussed in the main text.

- The mutations in Figure 4 confirm the importance of those residues involved in terms of catalysis, but not necessarily in terms of facilitating the open-to-closed conformational change. Authors can consider running MD simulations of these mutants and see if any changes in conformational dynamics.

- The crystallography data table contains some errors. Rmerge >100% for outer most shell is wrong. Rwork and Rkick for the AtIA-gl data set are both too high at 2.28Å resolution. The authors commented that the poor quality of the AtIA-gl crystal is responsible for the high Rfactor but all other statistics for this data look normal (i.e. completion, intensity and redundancy etc.)

Ljubljana, February 5th, 2019

Dr. Dušan Turk

Structural Biology Group

Dear Referees,

We would like to thank you for your time reviewing our work and providing valuable comments that enabled us to improve the overall quality of our manuscript. We took care to make some of its parts clearer as well as present the biological implications of our work in a clearer way. We have followed all your recommendations when preparing the new updated version.

1. A weak point in this manuscript is lack of the in vivo implications for this system, something that limits the impact of the present study.

Our work furthers understanding of the molecular mechanisms of controlling the local rigidity of the cell wall, potentially via such conformational changes as observed for the two studied proteins. Some of the previous studies have shown that the change in rigidity of the peptidoglycan is important in *S. aureus* cell division process. They are now properly cited.

Following the remark, we added text in the discussion section (line 300-309):

»Bailey et al. have shown that the newly formed peptidoglycan at the septum is stiffer and proposed that the subsequent drop in rigidity during division process is likely due to the peptidoglycan hydrolase activity⁴⁸. This coincides with AtIA localization at the septal region and its proposed role in cell division^{14,21} and their ability to cleave highly cross linked *S. aureus* peptidoglycan¹⁹. Superimposition of other 8 structures from GH73 family deposited in PDB provided no evidence that any of them were crystallized in other than the closed form (Supplementary Fig. 1). Hence, the insight into AtIA and SagB may shed a new light on the specific process of synthesis of new peptidoglycan in *S. aureus*, which is inserted only during cell division at the septum⁴⁹. Taken together, we can speculate that SagB and AtIA-gl have a role in cell division which involves reducing the rigidity of the peptidoglycan. «

2. Issues with crystallographic table, Rmerge, Rkick and AtIA-gl crystal structure quality.

We apologise for the errors in the crystallographic table as well as the missing data and thank the referees for directing our attention to them; something must have gone wrong with the multiple versions of the table while preparing the manuscript. We are sorry that the validation reports were not forwarded to the reviewers, they were included with initial submission. We are adding them again. Note that the deposits are already opened in PDB. Electron density images are now provided added. When looking carefully, one can see that the SagB map is of high quality as the Phe residue at the bottom actually has a hole in the middle of the ring. Nevertheless, as both referees indicated potential problems, we checked the structure determination procedures again and decided to redeposit the AtIA-gl structure at lower resolution than initially.

Table 1: Data collection and refinement statistics.

	AtlA-gl	SagB	SagB_SeMet_remote
Data collection			
Space group	P 64 2 2 (181)	P 61 2 2 (178)	P 61 2 2 (178)
Cell dimensions			
a, b, c (Å)	110.17, 110.17, 92.39	151.77, 151.77, 122.51	152.78, 152.78, 123.83
α, β, γ (°)	90, 90, 120	90, 90, 120	90, 90, 120
Wavelength (Å)	0.9184	0.8943	0.9770
Resolution (Å)	42.39-2.28 (2.41-2.28)	47.66-2.03 (2.15-2.03)	48.10-2.19 (2.32-2.19)
R_{measured} (%)	9.1 (162.6)	9.1 (105.0)	17.5 (192.7)
$I / \sigma I$	25.31 (1.96)	27.07 (2.02)	17.06 (1.83)
Completeness (%)	99.8 (99.3)	99.7 (98.5)	99.8 (99.0)
Multiplicity	21.1	18.3	21.2
CC(1/2)	100.0 (63.1)	99.9 (70.4)	99.9 (62.8)
ISa	24.95	31.33	24.85
Refinement			
Resolution (Å)	36.06-2.50 (2.54-2.50)	19.95-2.03 (2.07-2.03)	
No. reflections	11943 (578)	53814 (2544)	
$R_{\text{work}}/R_{\text{kick}}^1$	0.295/0.334	0.212/0.220	
No. atoms			
Protein	2261	4046	
Ligand/ion	3	107	
Water	162	548	
B-factors			
Protein	68.1	40.2	
Ligand/ion	68.1	66.6	
Water	66.6	66.2	
R.m.s. deviations			
Bond lengths (Å)	0.79	0.78	
Bond angles (°)	1.08	0.78	

¹ The idea behind R_{kick} is similar to the R_{free}, however, it is calculated from the structure factors of the kicked model against WORK set of data and not from structure factors of unperturbed model against TEST part of data. This corrects the conceptual problem of R_{free}. Namely, the claim that TEST set used in cross validation maximum likelihood refinement target function as source of structure independent information is false because errors in models under refinement are not randomly distributed⁵⁵.

Reply Figure 1: Stereo image of AtIA-gl 2mFo-DFc electron density Free Kick weighted map in the active site contoured at 1.6 sigma.

Reply Figure 2: Stereo image of SagB 2mFo-DFc electron density Free Kick weighted map in the active site contoured at 1.3 sigma.

The R_{measured} in the highest resolution shell are indeed over 100%. Yet, all data processing was done at the BESSY synchrotron with their XDSApp, which follow the latest recommendations, as described in Assessing and maximizing data quality in macromolecular crystallography by Karplus and Diederichs, *Curr Opin Struct Biol.* 2015 Oct; 34: 60–68.:

»Flawed indicators (R_{merge}) that have been replaced by R_{meas} . We recommend these be removed from all data reduction software. R_{meas} (=Rrim): Multiplicity independent replacement of R_{merge} and R_{sym} ; Useful for assessing space group symmetry and isomorphism of multiple data sets; Should play no role in determining resolution cutoff. ... In our view, no indicators other than CC1/2 should influence the high-resolution cutoff decisions for data processing. ... Until recently, a common and recommended practice has been truncating data at the resolution at which R_{meas} remains below ~60% and $\langle 1/\sigma \rangle_{\text{mrgd}}$ is ~2 or higher. Our report introducing CC1/2 also introduced paired refinement tests and showed that, for our test cases, including data out to a CC1/2 value of between 0.1 and 0.2

led to an improved refined model even though the data at that resolution had $R_{\text{meas}} \sim 450\%$ and $\langle I/\sigma \rangle_{\text{mrgd}} \sim 0.3$. We also showed that these weak data improved the quality of difference maps. This reinforced earlier evidence for the value in refinement of data having $\langle I/\sigma \rangle_{\text{mrgd}} \sim 0.5$. The damage caused by using an $R_{\text{meas}} \sim 60\%$ cutoff criterion grows with increasing multiplicity, because the excluded data have a higher and higher $\langle I/\sigma \rangle_{\text{mrgd}}$ which suggested us to follow these recommendations.

Explanation regarding the R_{kick} : The idea behind R_{kick} is similar to R_{free} , however, it is calculated from the structure factors of the kicked model against the WORK set of data and not from the structure factors of unperturbed model against the TEST part of data. This corrects the conceptual problem of R_{free} . Namely, the claim that TEST subset of structure factors used in cross validation maximum likelihood refinement target function as source of structure independent information is false as errors in models under refinement are not randomly distributed. In short, the TEST set of structure factors depends on the structure, due to the dependence of structure factors from the chemical energy terms used in refinement. Regularized atomic model spreads errors throughout the structure which essentially breaks the assumption of random distribution of errors – an essential condition for justified use of maximum-likelihood formalism. To break apart from this dependence, the whole structure is kicked (all atoms are displaced randomly along each coordinate by the positional error estimate) at each step of refinement. The kicked model is used to calculate the now independent set of structure factors which are then used to estimate the phase errors during comparison with the unperturbed WORK set. Thereby the use of the maximum likelihood target function in refinement became justified. As a side effect one can use all structure factors for the WORK as well as for TEST sets. Moreover, convergence of refinement does not suffer from the OFF target effect of the WORK set lacking the TEST portion of data and leads to more accurate structures. Longer description can be found in the paper *PRAŽNIKAR, Jure, TURK, Dušan. (2014) Free kick instead of cross-validation in maximum-likelihood refinement of macromolecular crystal structures. Acta crystallographica. D, Biological crystallography, ISSN 1399-0047, 70, 12, 3124-3134.*

Regarding the AtIA-gl structure quality, Rfactors are indeed high. We applied several approaches to reduce them. In the end we decided that the solution that resulted in electron density map which resolved most details would be best. We were able to prepare only a single crystal of AtIA-gl, which was not of great quality and collected diffraction data from it. (See attached Reply Figure 3) The maximum intensities on diffraction images were up to 1600, and often only up to 700 - 800. Thus the data were weak and so are the resulting accuracies of reflections. In addition, there is a substantial intensity drop in the resolution range from 5.3 to 7.0 Å not evident unless the data are divided into a large number of shells nor it can be remedied by the bulk solvent correction. The diffraction data analysis nor refinement and crystal packing indicated presence of twinning or anisotropy. (The latter was less likely due to the small crystal size anyway.) The exploration of lower symmetry space groups did not prove insightful either. In addition to P6422 we refined the structure in P64 and P31 space groups including lower resolution. (Due to high noise resulting in bad electron density maps, the P1 data set did not make much sense.) When we refined the structure in the P31 space group with four molecules in the asymmetric unit it lacked resolution of the P6422 form – even though lower R-factors 0.27 and 0.31 for the WORK and the KICK set, respectively, were obtained. When we used the refined structure of the P31 space group composed of four molecules in the asymmetric unit and refined them against the P6422 space group data, the electron density map has lost on resolution giving R-factors 0.31/0.34. In P6422 space group we tried several resolution cutoffs. For example, lowering resolution of the P6422 data set to 3.0 Å lowered R-factors to 0.26/0.30, however, resolution of features in electron density maps suffered too. At 3.3 Å resolution cutoff R-factor is reduced for further to 0.24/0.28. Taking it all together, the P6422 space group was chosen as the best possible choice for final

refinement and deposition. To make the final decision about the resolution cutoff we have deposited the data set to "proteindiffraction.org" server and consulted our friends (Marcin Cymborowski working from the Wladek Minor lab at UVA in Charlottesville and Sandra Gabelli at John Hopkins University in Baltimore). The data set was processed with XDS, DIALS, and HKL 2000. No errors or wrong conclusions about the space group and twinning were exposed. The final consensual conclusion was that the reasonable resolution cutoff is at 2.5 Å (with $1/\sigma$ in the last resolution shell of 3.8). Consensual decision was also not to build the missing region from 128 to 140 due to poor density. We decided to rerefine and redeposit the structure, although the resulting structure does not contain any major changes. RMSD between the new and old deposit is 0.33 Å. Three peptide bonds were flipped and the flexible region 150-156 has different orientations of peptide bonds due to the Ramachandran plot restraints recently included in the program MAIN. The updated structure has three peptide bonds flipped and the orientations peptide bonds in the flexible region from 151 to 156 are different due to Ramachandran plot restraints recently introduced in MAIN. The updated structure was deposited in PDB and is replacing the current entry 6FXO.

Reply Figure 3: A diffraction image of AtIA-gl crystal.

3. Line 127. Please provide a supplemental figure showing conservation of such structural elements in GH73 family.

The comparison of the conserved regions has been added as Supplementary Fig. 1:

Supplementary Figure 1: The comparison of the positioning of the core helices in the GH73 family of enzymes. (A) All the core helices (in different shades of blue) of all the other GH73 structures, namely Acp (5WQW), Auto (3F17), StFlgJ (5DN4), SpFlgJ (3VWO), ScaH (5T1Q), LytB (4Q2W) and TM0633 (4QDN), are aligned to the core helices of AtIE (the rest of the structure is in white). (B) The core helices of SagB and AtIA-gl (in magenta shades) are superimposed to the core helices of AtIE (in blue). (The superimposition aligned the helices in the L-domain, whereas the SagB and AtIA-gl helices of the L-domain are shifted relative to the AtIE helices of the R-domain.) The positioning of the core helices of all the other structures is very similar, while the positioning of the equivalent helices of SagB and AtIA-gl is visibly different.

4. In order to understand how these autolysins could act in vivo a better description of changes in the active site upon rearrangement should be provided.

Added text in the revised version (lines 181-192):

» To gain insights into the structural flexibility of the two proteins closed form models were constructed. Both were modeled using the AtIE structure as the template. Simple models were created by superimposition of each domain of SagB and AtIA-gl to corresponding domain of AtIE followed by minimization of the combined models (Figure 1). The models obtained by superimposition resulted only in minor side chain clashes: 24 non-hydrogen atoms from 4 residues of the L-domain and 23 non-hydrogen atoms from 5 residues of D- domains of AtIA-gl were closer than 2.2 Å (the distance used by PDB structure validation tools to indicate possible false covalent bond interactions), whereas for the SagB 30 non-hydrogen atoms from 5 residues of the L-domain and 34 non-hydrogen atoms from 5 residues of the D-domain were closer than 2.2 Å. These clashes were resolved by minimization. These simple models were further adjusted to match the chain trace of the α 12 helix in the AtIE template (marked with an orange rectangle in Figure 1, see Methods for further details). «

And the text in the methods we are referring to (lines 345-354):

» The closed form models of both structures were constructed in two steps. First, we created the models by superimposition of the R- and L-domains on the AtIE structure. L-domains contained

residues from sequence ranges from Q69 to K176 and L72 to S186 from AtIA-gl and SagB, respectively, whereas the R-domains contained the omitted parts. Then the broken link and side chain clashes were adjusted by energy minimization. Visual inspection of the superimposed models indicated the position of the pivoting point proximal to residues I89 in AtIE, L67 in AtIA-gl, and L106 in SagB. We called the generated models simple models. These two models were further modified by extension of the domain-connecting helices. The α 13 helix in the AtIA-gl closed form model from S175 to K191 and the α 15 helix in the SagB closed form model from P169 to S186 were rebuilt to match their chain traces to the α 12 helix in the AtIE structure. «

To provide better insight in the structural transitions Movies 1 and 2, depicting the transition from open to closed structure for both discussed structures, have been added to Supplementary files material.

Movie 1: The transition of AtIA-gl from the open to the closed conformation. Active glutamate (E116) is depicted in red and the substrate binding YATD (Y214-D217) region in blue.

Movie 2: The transition of SagB from the open to the closed conformation. Active glutamate (E121) is depicted in red and the substrate binding YATD (Y209-D212) region in blue.

5. Is this rearrangement expected for other GH73 family members?

Added text (lines 304-305):

»Superimposition of other 8 structures from GH73 family deposited in PDB provided no evidence that any of them was crystallized in the form other than closed (Supplementary Figure 1). «

6. There is insufficient introduction on AtIE and its structure in complex with (NAG-NAM)*3. So it is not easy to follow the definition of "open" and "closed" states for these enzymes. The author can consider adding a scheme that shows the stepwise transition from the "open" form observed in the apo structure to the "closed" form ready for catalysis.

Additional description is now provided. (lines 66-71):

»We recently determined the crystal structures of a homologous protein, autolysin E (AtIE) in complex with substrate fragments, which we used together with other available structural data on lysozymes to build a model of (NAG-NAM)₃ substrate binding to AtIE¹⁶. We have also analyzed the amino acids that are important for the reaction to transpire and identified E138 as the catalytic general acid and Y224-A225-S226-D227 as the motif responsible for substrate binding¹⁷. «

Movies depicting the conformational change have been added.

Movie 1: The transition of AtIA-gl from the open to the closed conformation. Active glutamate (E116) is depicted in red and the substrate binding YATD (Y214-D217) region in blue.

Movie 2: The transition of SagB from the open to the closed conformation. Active glutamate (E121) is depicted in red and the substrate binding YATD (Y209-D212) region in blue.

7. The description of modeling the "closed" structure of SagB and AtIA-gl by extending the domain-connecting helix is not clear. What extension was added to generate a model that is eventually stable in MD simulation? This may lead to model bias, which should be discussed in the main text.

The helix extension refers to the end of "Comparison of the AtIA-gl, SagB, and AtIE structures" Results section, where the differences in the helical region are described, in order to provide a reference for the changes we introduced by extending the helix in SagB and particularly in AtIA-gl (lines 132-136):

»In the AtIA-gl structure, the 4th turn of helix α 13 is stretched out compared to the slightly longer equivalent helix α 15 in SagB and the even longer helix α 12 in AtIE. The equivalent helix in the AtIE structure is distorted after the fourth turn with the aromatic residue Y201 slightly exposed. The α 15 helix in SagB is almost a whole turn shorter and shares the same minor widening in the last turn of the helix at position F185. «

To build the closed form, the chains of a few residues were thus folded to match the fold of the residues along the chain of the AtIE α 12 helix. As there are no insertions or deletions of residues in the three sequences, the template driven modeling appeared meaningful and was corroborated by the MD runs.

8. The mutations in Figure 4 confirm the importance of those residues involved in terms of catalysis, but not necessarily in terms of facilitating the open-to-closed conformational change. Authors can consider running MD simulations of these mutants and see if any changes in conformational dynamics.

Our conclusion from the MD simulations was: »Taken together, the molecular simulations confirmed the stability of the rebuilt closed forms and their complexes with substrate models. Furthermore, our analysis suggested that the interdomain conformational flexibility of AtIA-gl and SagB is accompanied by a partial helix-folding/unfolding mechanism. « We did not observe directly, whether the structural transition to closed form is facilitated by the substrate binding.

We however followed the suggestion and performed molecular dynamics simulation with the only amino acid that could be potentially considered to be involved in facilitating the open-to close conformational change, Y191. This residue is located in the L-domain connecting α -helix (magenta in Figure 2). As the mutation of equivalent residue also produced inactive AtIE, for which we do not have any indication that it does undergo a similar conformational change at this time, it is likely that it is involved in substrate binding. Other mutated residues are located at positions which are not directly connected to the structural elements involved in the conformational transition.

We constructed the model of Y191A mutant of the AtIA-gl in closed conformation with modified interconnecting α -helix and simulated this system in the 500 ns MD simulation. The influence on the opening motion was evaluated by monitoring the changes of the same L-D interdomain distance as in other simulations. As you can see from the interdomain temporal distance graph (see below) no significant influence on the selected distance was observed in comparison to the wild-type AtIA-gl model.

Reply Figure 4: Monitored L-D interdomain distances (same residues as defined in Supplementary Figure 4) for the AtIA-gl with extended helix in the closed conformation with bound substrate (blue) and its Y191A point mutant (black) during the 500 ns MD simulation.

9. Is there any indication if *S. aureus* autolysins recognize naked peptidoglycan or peptide stems are required? In this case, how peptide stem could be affected by the structural arrangement? Are crosslinked peptidoglycan strands substrate for these autolysins?

There is reported experimental evidence that both enzymes can cleave purified *S. aureus* peptidoglycan, which is highly cross-linked (see Ref 19). Thus both enzymes are expected to cleave glycan strands with peptidyl crosslinks. Furthermore, the compatibility of the peptide stems with the closed structures was examined by constructing the 3D models of both closed structures of AtIA-gl and SagB with glycan strand and additionally introduced the first three amino acids of the peptide stem (Supplementary Fig. 3). The resulting 3D models fit to the closed forms and thereby provide further evidence that the closing of the structures upon binding of peptidoglycan fragments is feasible.

Supplementary Fig. 3:

Supplementary Figure 3: 3D model of the substrate, that consists of glycan strand (in green) and added L-Ala-D-iso-Gln-L-Lys part of the peptide stem (in orange) bound to the closed models of the structures. The peptide stems were modelled onto the glycan strand and subsequently minimized. It is evident that the presence of the peptide stems allows the binding of the substrate as well as it does not hinder the closing of the enzymes.

Added text (lines 211-213):

»To show that also the closed forms of AtIA-gl and SagB can accommodate peptidyl links attached to polysaccharides we built the models using the bound (NAG-NAM)₃ substrate with added L-Ala-D-iso-Gln-L-Lys part of the peptide stem (Supplement Figure 3). «

Added text (lines 302-303):

»This coincides with AtIA localization at the septal region and its proposed role in cell division^{14,21} and their ability to cleave highly cross-linked *S. aureus* peptidoglycan¹⁹. «

10. Line 146. “binding geometry previously established for the AtIE structure” please provide reference.

Reference No. 16. (Mihelič et al., *IUCrJ*, 2017, 185-198) has been provided.

11. Line 151-152: “Figure 3” is doubled

This has been corrected.

As a result of these changes we feel that the manuscript has been improved considerably and hope that it is now ready for publication. Below we supply the list of changes to the manuscript made according to your remarks.

REVIEWERS' COMMENTS:

Reviewer #1 (Remarks to the Author):

I thank the authors for properly answer all my previous comments and concerns. The manuscript has improved in quality and could be published in its present form.

Reviewer #2 (Remarks to the Author):

The authors fully addressed the first three major concerns I raised in the first round of review. They also addressed part of the fourth concern, i.e. the crystallography data table. However, there are still two minor concerns regarding the data table.

- The authors explained about the meaning of R_{pick} and argued for its superiority over R_{free} . This point is debatable. Given the universal acceptance of R_{free} , the authors should calculate R_{free} and report it along with R_{pick} .

- The authors seemed to use $I/\sigma I > 2$ as the criteria for resolution cutoff in their data sets. However, R_{merge} is close to 100% for the outer most shell for all three data sets. Such high R_{merge} would lead to higher R_{work}/R_{free} after refinement. It is well accepted that the criteria for resolution cutoff should consider both $I/\sigma I$ and R_{merge} for the outer most shell. For data sets collected at synchrotron sources, the criteria of $I/\sigma I > 5$ and $R_{merge} < 50\%$ can be taken as a good practical reference.

Ljubljana, March 9th, 2020

Dr. Dušan Turk
zstructural Biology Group

Dear Editor,

Below is a detailed response to the requests of the reviewer #2:

Sincerely,
Dusan Turk

A) Rfree/Rkick

“The authors explained about the meaning of Rpick and argued for its superiority over Rfree. This point is debatable. Given the universal acceptance of Rfree, the authors should calculate Rfree and report it along with Rpick.”

In the cited paper Praznikar et al., 2014, we suggested that refinement does not need to omit part of data, called TEST, set for the Maximum likelihood Rkick target approach. We have not used the TEST set in refinement of our structures since then. By today we've published several structures using 100% of data for the WORK as well as for the TEST set and such concerns were not raised. Hence, we can state that Rfree is not exclusively applied.

The suggestion to calculate Rfree from a structure refined by Free kick maximum likelihood target essentially does not make sense, because no part of measured reflections were left out of refinement, which is a mandatory condition for the cross validation method to be applied. Hence, Rfactor calculated for any portion of data would be close to Rwork – as for example shown in the validation report. In order to calculate Rfree we would have to choose a TEST set and then rerefine and in part rebuild the structure using cross validation approach (solvent region mostly). However, not knowing which random TEST to choose, the question arises how many Rfree values should we calculate and how many refinements should we perform? 1, 10, 100, 1000 or $N! / (N \cdot 0.95)^{N-1}$, where N represents all reflections and $N \cdot 0.95$ represents 0.95% reflections used in WORK set. (The factorial is a number of possible TEST set choices which is impossible to calculate.) Which, if any, would be the correct choice? This question addresses the core of the problem, which we analyzed in our 2014 paper in which we explained that the »cross validation« concept has several major flaws. First, the TEST structure factors are not really free of uncorrelated errors in the model, but are contaminated with them. The contamination takes place during refinement which optimizes fit of the model to chemical and x-ray energy terms. Second, omission of data from refinement results in off target effects. Third, in map calculations following refinement crystallographers are only pretending that their structures are not contaminated with the TEST set, yet not only TEST set reflections, but also model structure factors of unmeasured reflections are included in the map

calculations. Hence, during model building stage models sees all the data. And additionally, as pointed out during discussion during the European crystallographic meeting, reflections of twinned data and NCS structures have even higher inter dependency. In order to avoid these unsatisfied conditions/assumptions on which the “cross validation” approach relies, the “free kick” approach was developed. As one can read in our paper, we refined complete and partial structures and those containing errors against resolution truncated and complete data sets. Analysis showed that the Rkick approach delivers more accurate structures than the “cross correlation” approach using Rfree. Furthermore, we showed that different random choices of TEST sets result in different structures as consequence of different off target effects. The analysis further showed that correlation between phase error and Rfree gap among structures refined against these different TEST sets does not exist. Hence, we showed that Rfree gap is a statistically meaningless number. As the only paper in that issue our work received attention of the editorial board and Alexandre Urzhumtsev wrote a scientific commentary “Free or not to free” (2014, Acta Cryst D, p 3088-3089 <https://doi.org/10.1107/S1399004714025413>) and an excerpt from the paper:

“...the current idea by Pražnikar & Turk presents a significant methodological step forward. In fact their idea has consequences going much further. Excluding a part of the reflections from refinement means a loss of experimental information. Moreover, Stest reflections have to be excluded not only from refinement but also, in principle, from the calculation of the Fourier maps used for model rebuilding; these corrupted maps may be another source of eventual model error (or of a bias in Rfree if reflections are not excluded). Pražnikar & Turk claim that further development of alternative validation techniques will eventually make calculation of Rfree unnecessary, being substituted by other approaches. This would mean the possibility of refinement against a full set of collected structure-factor amplitudes using the accurate and robust ML-approach suggested in their article.”

I have presented this work at a number of seminars in US and Europe in addition to the talks at European and American Crystallographic Association meetings in 2015. My talks were well accepted and I received positive comments addressing our work as original and out of the box. Later on, the results of this work were once disputed on the CCP4 bulletin board by Axel Brunger and Paul Adams. (Axel Brunger is the inventor of Rfree approach.) We, Jure Pražnikar and myself, responded and clarified the issues that were not raised again. **Hence, the “debatable” issue raised by the reviewer #2 stopped there.** Sadly, the Rkick approach was not implemented in major refinement programs such as PHENIX and REFMAC. In 2017 I’ve published a review “Boxes of Model Building and Visualization” (in Series: Methods In Molecular Biology: Protein Crystallography, Overview | DOI: 10.1007/978-1-4939-7000-1_21), where I’ve discussed that Rfree is a result of wishful thinking - a desire to have a single number that would say it all. There one can also read that it was not the first time in crystallographic community that an idea (iterative refinement) was at first not accepted by the community, but later, when the time was right, nevertheless accepted generally.

Saying this, I conclude that calculation of an Rfree in the context of this work is meaningless. I can however make a promise that we will address the Rfree gap religion again in a paper, which we will prepare in a year or so and submit it to an appropriate journal.

B) Resolution cutoff

“The authored seemed to use $I/\text{SigI} > 2$ as the criteria for resolution cutoff in their data sets. However, Rmerge is close to 100% for the outer most shell for all three data sets. Such high Rmerge would lead to higher Rwork/Rfree after refinement. It is well accepted that the criteria for resolution cutoff should consider both I/SigI and Rmerge for the outer most shell. For data sets collected at synchrotron sources, the criteria of $I/\text{SigI} > 5$ and Rmerge $< 50\%$ can be taken as a good practical reference.”

We have already responded to the critical arguments in the revised version and accompanying letter and included citation of appropriate scientific literature. (We also cut the resolution in the case of AtIA-gl structure, to accommodate the reviewers comments.) So, I've decided to ask colleagues from the crystallographic community for their opinion about the “good practical reference”. Several members of the crystallographic community responded to my mail to the CCP4 bulletin board mailing list. The responses ranged from the opinion that the specified criterium has been considered obsolete for a number of years, someone shared a view that if the “good practical reference” criteria was used and the unmatched structures deleted from PDB, the number of PDB deposits would be decimated, to the suggestion to request a different reviewer.